# Set the tone: Trustworthy and dominant novel voices classification using explicit judgement and machine learning techniques

**Cyrielle Chappuis** ⓘ *, **Didier Grandjean**

Neuroscience of Emotion and Affective Dynamics Lab, Faculty of Psychology and Educational Sciences and Swiss Center for Affective Sciences, University of Geneva, Geneva, Switzerland

\* cyrielle.chappuis@etu.unige.ch

**Data Availability Statement:** All relevant data are within the manuscript and its Supporting information files.

**Funding:** The authors received no specific funding for this work.

## Abstract

Prior research has established that valence-trustworthiness and power-dominance are the two main dimensions of voice evaluation at zero-acquaintance. These impressions shape many of our interactions and high-impact decisions, so it is crucial for many domains to understand this dynamic. Yet, the relationship between acoustical properties of novel voices and personality/attitudinal traits attributions remains poorly understood. The fundamental problem of understanding vocal impressions and relative decision-making is linked to the complex nature of the acoustical properties in voices. In order to disentangle this relationship, this study extends the line of research on the acoustical bases of vocal impressions in two ways. First, by attempting to replicate previous finding on the bi-dimensional nature of first impressions: using personality judgements and establishing a correspondence between acoustics and voice-first-impression (VFI) dimensions relative to sex (Study 1). Second (Study 2), by exploring the non-linear relationships between acoustical parameters and VFI by the means of machine learning models. In accordance with literature, a bi-dimensional projection comprising valence-trustworthiness and power-dominance evaluations is found to explain 80% of the VFI. In study 1, brighter (high center of gravity), smoother (low shimmers), and louder (high minimum intensity) voices reflected trustworthiness, while vocal roughness (harmonic to noise-ratio), energy in the high frequencies (Energy3250), pitch (Quantile 1, Quantile 5) and lower range of pitch values reflected dominance. In study 2, above chance classification of vocal profiles was achieved by both Support Vector Machine (77.78%) and Random-Forest (Out-Of-Bag = 36.14) classifiers, generally confirming that machine learning algorithms could predict first impressions from voices. Hence results support a bi-dimensional structure to VFI, emphasize the usefulness of machine learning techniques in understanding vocal impressions, and shed light on the influence of sex on VFI formation.

## Introduction

In 1935, social psychologists Cantril and Allport [1] published a book entitled "the psychology of radio". In one of the first occurrence of a scientific examination of recorded voices, the opus

**Competing interests:** The authors have declared that no competing interests exist.

was partially dedicated to describing how radio speakers were perceived by their audience. The radio era launched a new mode of communication based on auditory input only. Broadcast was a way to reach out to audiences in an entirely novel medium. Similar to ancient Greece orators' concerns, questions were raised about how radio speakers could persuade their audience, and to what extent. Since then, the number of communications happening through voice alone have only increased. Encounters with novel voices, speakers we have never seen nor met, is extremely frequent. Despite the lack of acquaintance and sometimes piece meal exposure, rich representations of newly met individuals are formed. As Cantril and Allport [1] point out:

> *"Voices have a way of provoking curiosity, of arousing a train of imagination that will conjure up a substantial and congruent personality to support and harmonize with the disembodied voice. Often the listener goes no further than deciding half- consciously that he either likes or dislikes the voice, but sometimes he gives more definite judgments, a "character reading" of the speaker, as it were, based upon voice alone."*

> *(p.109)*

How are a wide range of traits inferred without prior interaction? How is the rich construct of one's personality built?

First impressions are formed on various sources including faces [2–9], body language [10–12], and voice [13–17]. In 1946, Asch pointed out that upon meeting a new person, a unified impression was quickly and effortlessly formed from the start despite the large number and heterogeneity of trait characteristics to be evaluated [18]. Asch's work produced several important results which constituted the initial data for social cognition: the traits are processed in an organized and structured way, contribute differently (difference between "central" and "peripheral" traits), and dynamically interact to project a unified impression [18, 19].

First impressions apply to evaluation of many enduring trait characteristics such as trustworthiness [9, 20], competence [21–23], integrity [24], dominance [25–28], attractiveness [9, 29], likeability [9], warmth [23], and aggressiveness [9]. These traits can be summarized on fundamental dimensions of evaluation, of which two consistently emerge in literature [30]. Oosterhof and Todorov [5] examined the structure of face evaluation by identifying traits spontaneously used by participants to describe neutral faces, measuring judgments on these traits, and using a principal component analysis to reduce data dimensionality. Their result revealed that face first impression formation could be summarized on a two-dimensional space. The first principal component was interpreted as a "valence" evaluation because of its positive association to all positive traits, and negative association to all negative ones. This evaluative dimension is described by the authors as the result of perceived facial cues as signals associated to adaptive tendencies (approach or avoid). The second component was labelled "dominance" for its strong association with dominance and aggressiveness judgments. According to the authors, this evaluative dimension represents perception of physical strength and weakness in facial cues. Several overgeneralization theories have been proposed to explain in an adaptive perspective how facial cues determine specific trait inferences [31] and provide a rationale for finding somewhat consensual impressions across perceivers. The common key reasoning behind these possible explanations is that trait inference reflects a repeated exposure to a type of facial disposition associated with a social outcome of adaptive significance. Such facial dispositions can reflect low fitness (*anomalous face overgeneralization*, [32]), maturity (*baby face overgeneralization*, [33]), familiarity [34] and emotions [35]. For instance, a display of positive facial expression (ex. happiness) activates a tendency to approach the source and is

associated to positive social traits, while a negative one (ex. anger) signals hostility and prompts a tendency to avoid the source of possible harm.

## Vocal impressions

Voice constitutes a valuable source of social information. In groups, vocal communication has been found to maintain social cohesion, reduce food competition or predation risks [36]. Similar to face evaluation, vocal impressions can be represented in a two-dimensional space of warmth by dominance judgements [37, 38]. In a study where participants were asked to listen to vocal samples (recording of the sentence "hello, hello") before making personality judgments and occupational category guesses, Yamada and colleagues [39], uncovered a three dimensional appraisal of target voices. These three dimensions were social desirability, activity, and intelligence. The first one, social desirability, had high loadings of kindness, conscientiousness, safety, honesty, favorableness, trustworthiness, and warmth traits. The second one, activity, accounted better for extroversion, physical activity, cheerfulness, optimism, and humor. Finally, the intelligence dimension accounted for adjectives such as neat, bright, intelligent, stable, and mature. Yamada and colleagues' [39] results also showed a consistency in use of vocal cues to make personality inferences, and categorize target persons on specific professions. In order to identify the underlying dimensions of voice evaluation, McAleer, Todorov and Belin [13] presented participants with recordings of the single word "hello" and asked them to rate the speakers on ten personality scales. For both male and female voices, a two-dimensional solution, in line with well-established dimensional models of trait inference [9, 40], was found to account for most of the variance in the evaluation data. The first dimension, gathered positive loadings of every traits except for aggressiveness. It approximated the "valence" dimension described in prior models of facial impression [5]. The second dimension mainly represented judgments of aggressiveness, attractiveness, confidence and dominance, with negative loadings on likeability, trustworthiness, and warmth. As such, it approximated the "dominance" dimension described by Oosterhof and Todorov [5]. Interesting for the purposes of the present experiment, valence was positively associated with a rising pitch and harmonic to noise-ration (HNR) in females, while, quite surprisingly, a positive relationship between pitch, HNR and valence was found for male speakers. For both female and male voices, formant dispersion was found to explain perception of dominance. This study successfully provided strong evidence of an organized vocal impression in a two-dimensional space. However, it also raises several questions.

First, is vocal first impression related to the speaker's personality? Indeed, speakers could have learned to use their expressive cues to project a socially desirable trait and enhancing the recognition of the trait by the receiver [41]. Extraversion has notably obtained the most compelling evidence for agreement across literature [15, 42–44]. For example, extraversion rating at zero-acquaintance has been correlated to the perception of a large number of physical attributes such as "soft-powerful" voices [45]. Perceivers happen to be sometimes accurate judges of the speaker's self-assessed personality. Berry [37] reported significant associations between perception of vocal power (akin to the "dominance" dimension in the valence-dominance model) and womens' self-assessment of aggression and power (but not assertiveness). In turn, female speakers who rated themselves as high on social closeness were also perceived as higher on the warmth dimension.

Second, as mentioned earlier, phenotypic substrates influencing acoustical features could play a strong role on vocal impression through the percepts related to those acoustical features modulations, e.g. local pitch and amplitude variations have been suggested to be of importance. In McAleer, Todorov and Belins' [13] study, jitter and shimmer were not found to play

a role on person perception, although they are strong vocal quality cues. Jitter and shimmers are best measured on sustained vowels. Hence, could these results be replicated using longer excerpts, allowing more acoustical data variations to be obtained?

Thirdly, is there a complex interaction between the features and VFI? It has been demonstrated that features interact dynamically, as for instance lower f0 has been found to improve the perceptual salience of formants [46]?

There is consistent evidence in literature that VFI reflect the heuristic assumption that voice pitch is proportional to the size or formidability of the signaler: i.e. that larger individuals produce deeper, lower sounds [47]. Morton's [48] motivational-structural rules hypothesis suggested that signallers and receivers exploit a relationship between the acoustical structures of the vocalizations and the signaler's motivations or emotional state. Specifically, rough, aperiodic low frequency sounds are used to signal (and infer) hostility, while at the opposite, tonal and high frequency sounds are used in friendly and appeasing situations. This hypothesis relies on the assumption that fundamental frequency is inversely related to body size, and by extension, strength and likelihood to prevail in a competition. Vocal roughness also contributes to this impression, because larger vocal folds are more likely to produce aperiodicity [49]. In human vocalizations, lower fundamental frequency is thus perceived as "more dominant" and "self-confident" [50].

Therefore, in a functional perspective, it would be reasonable to expect that the structure of first impressions from voices reflects, to some extent, the frequency code and relative body size projection principle. In other words, the adaptive value of responding appropriately to the inferred fearfulness or attractiveness of a voice and thus avoiding or approaching the individual would be related to a tendency to process vocalization cues reflecting body size and infer analogous traits. Body size has been found to be perceptually related to formant dispersion [46, 51–54] and mean f0 [54–56]. Some studies have addressed the direction of this link: for instance, lower fundamental frequency would be associated to capability or dominance perception, because of the association between hormone levels and vocal fold's mass [57, 58], competition [59, 60], and large body size [47]. Low pitch in both females and males has also been linked to high ratings of leadership, dominance, and trustworthiness in a study of leadership selection [61]. In regard to the baby overgeneralization hypothesis, perceivers use indicators of vocal maturity to make trait inferences on the weakness, incompetence, and warmth dimensions [38]. Specifically, a voice with perceived acoustical characteristics resembling one of a child tend to be judged as weaker, more incompetent than mature ones [37, 38], and to receive higher ratings on the warmth dimension [37].

## The present study

Thus, there are three aims to the present study. First, we attempt to replicate, to a certain extent, McAleer, Todorov and Belin's [13] results. In accordance with previous literature, we expect VFI to be organized, core-centered, and to revolve around valence and dominance dimensions. Second, in a brunswikian perspective [62, 63] we compare receiver's evaluation to the signalers' own personality characteristics. We expect to find correspondences between personality factors of the Big Five and dimensions of person evaluation based on voices. Thirdly, we also investigated the relationship between acoustics and impressions. Insofar the question of the structure of impression formation from voices has not been addressed in light of its adaptive significance, or more specifically, taking into account the contingencies between the dimensions of inference and the social concerns to which such a process provides a solution. We argue that if vocal impression formation solves an evolutionary problem, then specific combinations of vocal characteristics hold informative value that can be parsed and

categorized into attitudinal dimensions. Thus, we apply machine learning algorithms to determine whether some trait inferences can be used to train an automated classifier, and to identity how vocal cues interact to provide the best ground for guessed classification. In light of the data on vocalizations, we expect that different acoustical properties make up for the evaluation of orthogonal dimensions, and that specifically, acoustical features associated to body size and relative shape would be strong indicators of traits. The goal of this last analysis is twofold: understand non-linear interactions between acoustical features, and test the accuracy of a trained classifier on attitudinal voice categorization.

## Study 1: First impression evaluation and acoustical underpinnings

### Method

**Stimuli preparation.**   Forty-eight native French speakers between 18 and 35 years old (20F/20M, M = 22.47, SD = 2.61) were recruited at the University of Geneva, out of which eight participants whose recordings failed were excluded from the study. The recordings took place in a sound-proof laboratory. Speakers were equipped with a head-worn microphone (HSP4, Sennheiser), and were instructed to stand straight in front of the display screen during the recordings. Each speaker could then silently read the target sentences on the screen before repeating them out loud. A 30-seconds delay was inserted between the silent reading and the recording phases in order to let the speaker rehearse and avoid a reading-like voice. Speaker subsequently recorded the four sentences displayed in random order on the screen (see S1 Text in S1 File for original translation): « I think I'm a worthy candidate for this position », « I am qualified to carry out this task », « I have the required competencies to carry out this task », « I think I have required competencies for that position. Long utterances stimuli were chosen with the aim of analyzing language dynamics such as speech rate. All sounds were recorded in a.wav format, as 32-bit mono files at a sampling frequency of 44.1-kHz, and then normalized for loudness (RMS normalization) using MIRToolbox [64]. After recording the sentences, each speaker was given a NEO-FFI and a DS36 test. The NEO-FFI is the shortened version of the Revised NEO personality inventory (NEO PI-R) [65], which is composed of 60 items (12 items per personality domain). The NEO-FFI can be completed within 15 minutes, and is based of the five personality factors previously operationalized by the NEO-PI test: neuroticism, extraversion, openness to experience, agreeableness, and conscientiousness.

**Acoustic measures.**   For each of the 160 vocal stimuli ($M_{duration}$ = 2.87ms, $SD_{duration}$ = 0.44ms), 41 measures related to pitch, intensity, delivery and voice quality were extracted using PRAAT software [66] and the ProsodyPro tool [67]. Measures are detailed in S2 Text in S1 File.

**Participants.**   Forty-six participants took part in the evaluation (18M/22F, M = 22.75, SD = 3.13), and six were removed from the analyses because they recognized one or several speakers. All participants were French native speakers, and had normal hearing.

**Procedure.**   The experiment took place at the University of Geneva. In order for the assessment to be made in a reasonable amount of time, each participant was given 40 stimuli to assess instead of the total of 160. Stimuli presentation was counterbalanced to ensure that all stimuli were assessed by an equal number of participants, and that no more than two voices of the same gender or expressing the same utterance would be presented subsequently. The experiment was divided into ten blocks (with up to two minutes breaks in-between), one for each attitudinal dimension to assess. Hence, every participant went through all dimension conditions in a random order. For reproducibility purposes, all dimensions were based on McAleer, Todorov & Belin's [13] study, namely: Aggressiveness, Attractiveness, Competence, Confidence, Dominance, Likeability, Trustworthiness, and Warmth. A definition sheet was

given to each participant and available during the whole experiment. In total, each participant had 400 assessments to make. Protocols for this study were approved by the ethics committee of the University of Geneva. Prior to participation, an informed consent was signed by the participants, and a debriefing was held after.

## Results

In accordance with Scherer's [15, 68] adaptation of Brunswick's lens model, we provide analyses for the mechanisms influencing both distal cues at the speaker's level, the proximal cues and the attributions at the receiver's level. Firstly, we investigate the relationship between personality traits and voice. Secondly, we investigate personality and attitude inference based on voices.

**Personality results.** Participants took the NEO-FFI personality test which provided scores for five personality dimensions: neuroticism, extraversion, openness to experience, agreeableness, and consciousness. Pearsons correlation analyses showed that only neuroticism and openness to experience were significantly associated to the perceiver's attributions. Neuroticism correlated negatively with perception of dominance ($r(158) = -0.17$, $p < 0.05$), with a higher score of neuroticism being linked to a lower perception of dominance in voice. Openness to experience positively correlated with warmth ($r(158) = 0.23$, $p < 0.05$), but negatively correlated with both perceived aggressiveness ($r(158) = -0.18$, $p < 0.05$), and perceived dominance ($r(158) = -0.17$, $p < 0.05$).

**Impressions.** Our second research question addressed the structure of the impression, as we expected to find evaluative dimensions underlying trait inference. Firstly, we applied a principal component analysis (PCA) with Varimax rotations on all rated traits (Aggressiveness, Attractiveness, Competence, Confidence, Dominance, Likeability, Trustworthiness and Warmth) relative to sex (speakers). PCA have been found to be reliable measures of impression formation [69]. All traits taken together, the judgements showed a good internal consistency (Cronbach's $\alpha = 0.88$) as well as inter-rater reliability (see S1 Table in S1 File). Results (Table 1) show that a two-factor combination explains 80% of the total variance. The first component contributed for 49.2% of the total dataset's variance. Likeability, Attractiveness, Warmth, Competence, Trustworthiness, and Confidence, all had strong positive loadings on this component. A second component (PC2) was found to account for 30.8% of the dataset's variance, with strong positive associations to aggressiveness, dominance and confidence. Analyses by sex showed relatively similar patterns, with one difference on the factors

**Table 1. Weighted linear combinations of attitudinal traits shows individual contribution of all attitudes on the first and second principal components (PC) for all speakers, male speakers and female speakers, after PCA with Varimax rotation.**

| Attitude | All speakers | | Male speakers | | Female speakers | |
|---|---|---|---|---|---|---|
| | PC1 | PC2 | PC 1 | PC 2 | PC 1 | PC 2 |
| agreeableness | 0.89 | | 0.90 | -0.11 | 0.89 | |
| aggressiveness | -0.31 | 0.87 | -0.26 | 0.90 | -0.25 | 0.88 |
| attractiveness | 0.89 | | 0.91 | | 0.89 | |
| warmth | 0.78 | | 0.79 | | 0.74 | 0.22 |
| competence | 0.68 | 0.59 | 0.81 | 0.40 | 0.55 | 0.70 |
| trustworthiness | 0.85 | 0.33 | 0.90 | 0.14 | 0.78 | 0.44 |
| dominance | 0.33 | 0.88 | 0.54 | 0.76 | 0.24 | 0.91 |
| confidence | 0.60 | 0.70 | 0.77 | 0.48 | 0.42 | 0.84 |
| **Explained variance (%)** | 0.49 | 0.31 | 0.59 | 0.23 | 0.42 | 0.38 |
| **Cumulative variance (%)** | 0.49 | 0.80 | 0.59 | 0.81 | 0.42 | 0.80 |

representing competence and confidence which positively loaded on the dominance compo-
nent for female voices only (83.7% and 70.4% respectively, S1 Fig in S1 File). In male speakers,
competence and confidence weighed heavier impact to the valence component than domi-
nance. We tested the statistical difference between the two loading patterns by applying a Pro-
crustes analysis, generally used for investigating structural equivalence. This procedure allows
to compare the two solutions by rotating them and fitting them together. However, this differ-
ence did not significantly affect the two solutions, as Tucker's phi coefficients comparing the
factor structure in each sex in PC1 was 0.99 and 0.97 for the PC2.

**Acoustics.**    In order to understand what subset of acoustical feature underlie the dimen-
sions of impression, we applied a stepwise regression analysis. Results are summarized in S3
Table in S1 File.

The first component (PC1) predicted by linear combinations of spectral center of gravity ($t$
(149) = 2.56, p < .05), meaning that a more tonal timbre (aperiodicity) in the signal, the more
it was perceived as an indicator of positive valence by the listeners, shimmers (t(149) = 3.46,
p<0.01) and minimum intensity (t(149) = 2.11, p<0.05). Specifically, perception of valence in
females was positively associated to amplitude perturbation (shimmers: t(149) = 3.51, p < .01)
only. In male voices, valence evaluation was explained by a linear combination of energy in the
first quantile (quantile 1: t(149) = -4.72, p<0.001), minimum f0 (t(149) = 2.66, p<0.05), energy
in the fifth quantile (t(149) = 2.13, p<0.05), and marginally, energy below 1000Hz (t(149) =
-1.67, p<0.10). Perception of PC2 (dominance) was associated to formant dispersion (F1-F5: t
(149) = -1.99, p<0.05), energy in the 2750-3250Hz band (t(149) = 7.06, p<0.001), harmonicity
(HNR: t(149) = -4.05, p<0.001), and energy in the fourth quantile (quantile 4: t(149) = 3.38,
p<0.01). Specifically, perception of dominance in female voices was linearly predicted by har-
monicity (HNR: t(149) = -4.92, p<0.001), energy from 0 to 500Hz (t(149) = 4.24, p<0.001),
250Hz to 750Hz (t(149) = -3.18, p<0.01) and 3000 to 3500Hz (t(149) = 3.56, p<0.01, speech
rate (t(149) = 2.84, p<0.01), and marginally formant dispersion (F1F3: t(149) = 1.97, p = 0.05).
For male voices, energy below 500Hz (t(149) = -3.25, p<0.01) and energy bands in high fre-
quencies (3000-3500Hz: t(149) = 3.63, p<0.01; 3250-3750Hz: t(149) = -2.07, p<0.05) predicted
the perception of vocal dominance.

## Conclusion

In this study, we analyzed the structure of first impressions made from speaker's voices on sen-
tence-long inputs. As previously shown in research [13], we observe that attitudinal perception
from auditory cues is organized, and that a two dimensional solution can account for most of
the dataset's variance. This observation is in line with ecological perspectives of impression
formation [5], and with dimensions of social evaluations found in literature [9, 13, 23, 27, 62].
Organization of attitudes on both dimensions is merely similar between sexes, apart from a
small—but not sufficient to change the overall solution differences on competence and confi-
dence loadings. A possible explanation could be that perception of female and male's attitudes
often differ in literature, as the attribution of such traits can reveal different meanings or
expectations depending on sex [70–72]. Evaluation of the speaker's personality traits through
the scope of the Big Five inventory yields to the slight but significant relationship between neu-
roticism, openness to experience and vocal evaluations. The absence of significant correlation
vocal evaluations and extraversion was somewhat surprising, given that it often stands out as
the most robustly recognizable personality factor in studies [15, 43], and more generally seem
like an important trait in social interactions.

As expected, the acoustic analyses showed that different combinations of acoustical features
predicted the two dimensions. In regard to our expectations on the importance of fitness-

relevant parameters on trait inference, the results are mixed. On one hand, perception of valence was positively related to spectral center, shimmer and minimum intensity. Specifically, lower spectral center of gravity values were found be perceived as more untrustworthy. Perceptually, a high spectral center of gravity is related to a brighter sound, while a more diffuse center of gravity is perceived as "darker" [73]. Our results are somewhat opposite to Weiss and Burkhardt's (2010) who found that female and male darker voices were rated as more likable. In a functionalist perspective as argued by the authors, preference for both darker and brighter sounds would make sense, as they indicate different but relevant characteristics (such as youthfulness for brighter voices). Lower limits on the intensity range (minimum intensity) was also perceived as more untrustworthy. Shimmers are related to noise and breathiness in the vocal signal, leading to perceived roughness and breathiness. Because shimmers might be affected by physiological laryngeal tension (greater shimmers for low loudness), it is not so surprising to find that both features interact together. In female speakers, shimmers singly predicted perception of valence, meaning a higher degree of amplitude aperiodicity was perceived as more trustworthy. For male voices, shimmer were not predictive of valence nor dominance. There are two reasons shimmers might differently explain the perception of valence and dominance in females and males. First, shimmers have been found to be associated to vocal breathiness [74], so it is plausible that breathier voices are perceived as less threatening and more appeasing. Breathier voices have been found to be more attractive [47, 75] and likable [73]. Hence, even though Morton's [48] framework suggests that a rougher voice would be perceived as more aggressive, it is possible that breathiness counterbalances this effect. A second explanation for the relevance of shimmers in female voices but not males might come from sexual differences within types of phonation might, as females have been found to use softer voices (and hence, produce greater shimmers) than males when asked to produce normal phonation [76]. On the other hand, dominance was predicted by HNR, formant dispersion, energy in the fourth quantile and energy in the 3000-3500Hz band. Vocal HNR is an indicator of voice quality changes sometimes associated to perception of roughness [77, 78]. According to literature, a voice characterized by a high HNR is perceptually more sonorant and harmonic [74]. The presence of glottal noise measured by the HNR has also been associated to vocal aging, with decreasing values in elderly populations compared to younger ones [79]. Thus, a higher degree of harmonicity in voice was perceived as less dominant in the present experiment, and this was particularly the case for female voices. Voices were also perceived as more dominant with low values of formant dispersion (F1F5), a correlate of vocal tract length and shape. Narrower formant distances have been found to perceptually create more "resonance" in the voice, and to participate in the perception of vocal depth [80]. As a result of its physiological substrates, formant dispersion is greatly affected by sexual dimorphism and puberty [81]. It is somewhat surprising that it predicted dominance perception in female speakers but not males. For female voices, larger distances between formants (*Df* F1-F3) marginally (but not significantly) was associated to more perceived dominance. These results contrast with the ones reported on male speakers by Puts et al. [17] showing narrower formant distances lead to increased dominance perception, and that this link was stronger for physical dominance than social dominance. Perceived dominance in female voices was also positively associated to speech rate. This is in line with literature, where a faster delivery style has been associated to vocal charisma [82], competence (and significantly but differently, benevolence) [83], sarcasm [84], irritated, and resigned speech [85]. Energy also played an important role in dominance perception. For males, combinations of energy in the frequencies below 500Hz and in overlapping bands (3250-3750Hz, 3000-3500Hz) predicted perception of dominance, suggesting that energy peaks below 500Hz are associated to low dominance perception. This is consistent with results from Goudbeek and Scherer [86], who described this parameter's implication in potency/

control emotional appraisals, a dimension which shows some conceptual proximity with dominance.

In conclusion, this experiment successfully replicates the observed structures in literature from vocal impression [13], and sheds light on sex-specific patterns of impression. The results reveal different acoustic bases for valence and dominance perception, and emphasize the importance of voice quality (both in terms of variability and spectral balance) parameters on trait inference.

It is possible to argue that a major drawback in vocal impression literature is the study of single acoustic features or linear combinations in relationship to impression formation, which does not inform on the interactions between them. In order to address this caveat, we apply automated classifiers which allow the use of polynomial kernels to study potentially complex combinations between acoustic features.

## Study 2: First impression training using machine learning and acoustical feature utilization

The first study allowed the observation of a 2-D structure of impression formation revolving around two principal components: dominance and valence. However, factorial reduction analysis does not provide further information on the individual and interactive effect of each vocal cue in the component categorization. The main objective of this second study is to better characterize the link between acoustics and impression formation, while taking into account non-linear relationships and high dimensionality.

For this purpose, we determine which acoustical parameter in voice can be used to classify those into discrete categories. By training a model to compare vocal cues based on the attribution made by the listener, we expect to understand better which cues are the most important for such categorization, and how they relate. Automated classifiers have become increasingly popular for emotional prosody categorization as well as speech recognition [87–93], and in clinical diagnoses [94]. Automatic detection of vocal irritation and resignation was tested with a linear discriminant analysis on a corpus of 186 sounds. Compared to human performance, better than chance classification was observed, with close to human performances varying from 54.3% to 62.3% [85]. General discriminant analysis was also applied and compared to judges' ratings on portrayed emotions by Banse and Scherer [95], providing similar, and even better in some bases, correct recognition rates (for example for cold anger, and elation, but not interest or contempt). Despite being a fairly recent field of study, vocalizations have previously successfully been used as bases for automatic trait attribution. Mohammadi, Vinciarelli and Mortillaro [96] used a SVM with Radial Basis Function Kernel to assess a collection of 640 speech samples categorised by personality trait (Big Five inventory), achieving up to 75% accuracy. Those results provide encouraging ground for automatic attitude classification of voices on personality traits.

These experiments constitute a promising ground for the utilization of automated classifier in affective speech recognition. However, the study of first impressions differs from emotions. Unlike emotions, first impression formation is a construct formed on the receiver's side of the interaction, based on the integration of a constellation of traits, and thought to reflect stable properties of the speaker. Its description relates to the receiver and the utilization of distal cues to make inferences on enduring inter-individual differences. On the other hand, emotions reflect both intra-individual variations and inter-individual ones, not to the signaler's own motivations and intent. Accuracy is a recurring question in impression formation research, especially in the light of the ecological approach. However, it is somewhat limited to self-assessment of personality cues instead of portrayal of "personality" vocalizations. Indeed,

because it concerns personality judgement and that hypotheses suggest that distal cues are used as an approximation of fitness relevant cues, impression does not necessarily concerns transient and controllable aspects of speech that would be easily portrayed by actors. Hence, impressions do not benefit from a "true" or "correct" base on which to measure the recognition accuracy. Instead, it is possible to consider the level of agreement between judges as the correct label in order to determine if specific acoustical patterns provide ground for impression formation. We observed in the first study that a limited exposure to a novel voice was sufficient to form an impression of the speaker and to rate him/her on personality and attitudinal scales with high agreement among the perceivers (Cronbach's alphas:0.79–0.90). Similar to research on emotional voices, we suggest specific and complex relationships between acoustical cues and impressions. In other words, we assume that the recorded voices are generated by some probability distribution, and that there is a "correct" labelling function by which a label vector is paired with one or several feature values. We report and compare two supervised learning methods: a Support Vector Machine (SVM) and a Random Forest analysis (RF). This second study aims at determining which acoustical parameter play a role in trustworthiness and dominance catregorization, as well as to shed light on how their linear or non-linear combinations can be used to classify voices into discrete categories of impressions. For each analysis, categories for class labels are created based on the judgment scores obtained in Experiment 1.

## Results

**Support Vector Machine approach.** A SVM model is trained on three sets of analyses: the first one trained to discriminate between attitudinal dimensions, the second to discriminate between positive and negative valence voices, and the third trained to discriminate between high and low power voices as found in our data in the first evaluation study presented above. Support Vector Machine (SVM) models have the advantage of working well on small dataset, and the polynomial function allow researchers to test for feature's combinations effects on classification rather than single feature's influence [97]. SVMs allow classification by constructing an optimal linear decision boundary between two classes, based on the furthest distances from training data (hyperplane) [98]. The further the distance, the better the learning model performs at classifying speech signals from the training data into classes. The binary SVM is trained with an Error Correcting Output Code (ECOC) model in order to fit it to a multi-way classification and compute confusion matrices [99, 100]. All analyses are performed on Matlab 2018a software (the Mathworks) using the Statistics and Machine Learning Toolbox. We provide three sets of analyses: first, a classification of vocal profiles based on their associations to both valence and dominance dimensions, second, a classification based solely on the valence dimension and third, a classification based on the dominance dimension.

*Dimensional classification training.* In a first set of analyses, we train a SVM model to test whether two dimensions of vocal impression can be separated based on their acoustic features. Our voices were all evaluated on two dimensions approximating valence and dominance judgments. In order to distinguish vocal profiles, we created four class labels based on the polarized loadings returned by the PCA (See S1 Fig in S1 File): trustworthy and dominant voices (HTHD), untrustworthy and weak voices (LTLD), trustworthy and weak voices (HTLD), and untrustworthy and dominant voices (LTHD) Cut-off thresholds for class labels were loadings value above 0.5 (for "highly positive" loading on a dimension), and below 0.5 (for "highly negative" loading on a dimension), so a HTHD label refers to a voice sound saturating high on both dimensions, while HTLD characterizes a voice sound positive on the valence dimension but negative on the dominance one. Prior to training the SVM, a subset of features are selected

based on neighborhood component analysis (NCA), using feature weights and a relative tolerance threshold. A solution with thirteen acoustical features is found to lead to the best classification (by order of importance: COG, PE1000, Quantile4, Hammarberg index, CPP, SDIntensity, Medianf0, Energy3250, MeanF0, Energy750, Quantile3, H1H2, Shimmer), and the three best features' interaction can be visualized in S2 Fig in S1 File. A four degrees polynomial kernel is used in the SVM model as to obtain the best accuracy and predictive power while preventing overfitting (Kernel Function = "polynomial", "4"). Low-degrees kernels have been successfully tested and used in natural language processing and sometimes provide a highly accurate testing without overfitting [101–103]. Kernel scale parameter is set on "auto" (subsampling heuristically chosen, with a controlled random number generation). The box constraint value which penalizes misclassification and thus helps preventing overfitting is set at 6, and predictor data is standardized (Standardize = "1"). The k-fold cross validation parameter was set at 5, so the training set comprises four-fifth of the data (50 items) while one-fifth (13 items) is held out as testing set for each of all five experiments. Cross-validation solves the problem created by division of the dataset into a training set and a testing set by splitting the dataset into k-numbers of testing sets, subsequently fitting the model on all data, and finally computing the scores for number k of times. As a binary classifier, the SVM can solve multiclass classification by two techniques: one-versus-All or one-versus-one. We used the fitecoc function in MATLAB (MATLAB, version 2018a), which operates with a one-versus-one coding design, and provides a classification model for each of the four class of vocal profiles before combining them.SVM's overall accuracy (the rate of correct classification) was 77.78%. ECOC classifier shows that untrustworthy and dominant (LTHD) vocal profiles were correctly classified 100% of the time, whereas trustworthy/non-dominant profiles were classified with a 60% accuracy (Fig 1A). However, overall accuracy does not necessarily indicates how good a learning classifier is, especially when used with strongly unbalanced data such as often observed in real-world learning, or when the misclassification cost between classes is either unknown or unequal [104]. The area under Receiver Operating Characteristic (ROC) curve (Fig 2) can provide a more statistically discriminant classification than accuracy on real world datasets [105]. The (ROC) curve (Fig 2A) indicates the SVM's predictive ability with regards to its sensitivity (True positive rate) and specificity (True negative rate). Then, the ROC curve displays the rate of true positives compared to the rate of false positives. The Area Under the Curve (AUC) is the most frequently used measure of performance based on the ROC, it represents the probability that the classifier detects positive and negatives correctly. The larger the AUC is the closer it gets to an ideal classifier with perfect accuracy (AUC = 1). Results (Fig 2A) showed that the learning model had higher classification rates for trustworthy-dominant (AUC for HTHD = 089) and untrustworthy/non-dominant profiles (AUC for LTHD = 0.89). AUC values between 0.6 and 0.7 are considered sufficient, and above 0.7 are considered good [103].

*Dominance classification training*. The previous analysis shed light on the acoustic distinctiveness of vocal profiles. However, given the limited number of observation, it was decided to only compare profiles on the extreme side of each dimension (and neutral). This second analysis aims at examining closer the different realizations of the dominant vocal profiles, depending on its polarization (dominant, neutral, weak). Cut-off thresholds for class labels were loadings value above 0.7 for dominant voices, below -0.7 for weak, and between -0.3 and 0.3 for neutral. A subset of sixteen acoustical features were selected after NCA (by order of importance): SDf0, CPP, Energy3000, speech rate, MeanIntensity, Energy2250, Shimmer, Df(F1F5), MinIntensity, Quantile1, Jitter, Energy1500, HNR, f0max, Energy3500, Quantile5 (S3 Fig in S1 File). We use a one-degree polynomial kernel function (kernel scale = "auto"; Box constraint = "5"), with 7 cross-validation operations (K = 7; Training set = 79; Testing set = 20). The chosen

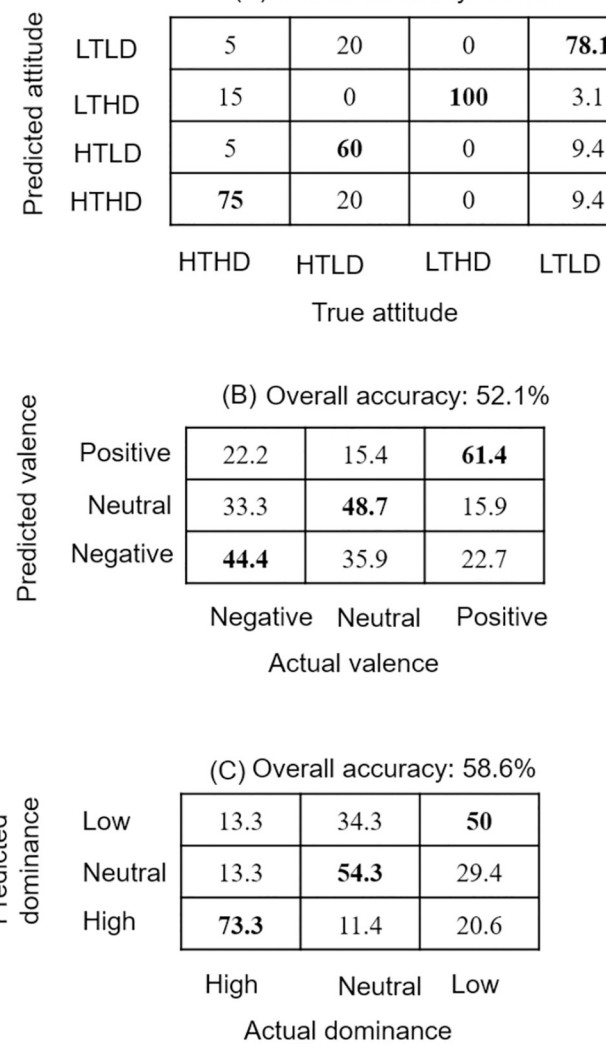

**Fig 1. Confusion matrices of p(true attitude | model prediction) for each SVM.** (A) Results of vocal classification for different attitudinal labels. with a 77.8% accuracy. From bottom left to top corner right are all the congruent classifications (HTHD: trustworthy/dominant, HTLD: trustworthy/non-dominant, LTHD: untrustworthy/dominant, LTLD: untrustworthy/non-dominant). True attitude represents the given class label, and predicted attitude is the assigned class label after supervised learning. (B) Results of vocal classification for valence (positive, negative or neutral). (C) Results of vocal classification for dominance (high, low or neutral), from bottom left corner to top right are all the correctly classified items (Low Dominance = 50%; neutral = 54.3%, high dominance = 73.3%).

K value allows a sufficient number of data to be held out as testing set without overfitting. Results show an overall classification accuracy of 58.59%. Probability of correctly classifying voices scoring high on dominance is 73.3, 54.3 for neutral voices and 50 for voices scoring low on dominance (Fig 1C). Predicted neutrality in voices has a 13.3% probability of being actual dominants and a 29.4% probability of being actually low on dominance.

*Valence classification training*. In order to investigate valence attribution in voice, we run the SVM on a dataset with data associated three class labels: negative valence, positive valence, and neutral valence (Kernel Function = "polynomial"; Polynomial order = "7"; Box constraint = "5"; cross validation K = "5"). Cut-off threshold for loadings is the same as for the dominance

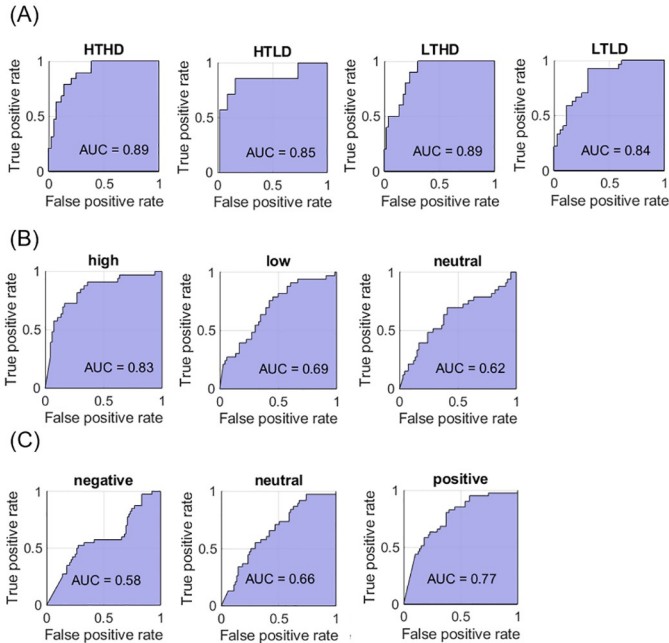

**Fig 2. Receiver Operating Characteristic (ROC) curves for bi-dimensional profiles (A), valence (B) and dominance (C) classification for Polynomial SVMs.**

classification, resulting in a dataset of 119 sounds (training data = 95; test data = 24). Feature selection based on NCA and a relative threshold highlights the importance of eight acoustical features used for the SVM (by order of importance): f0min, Df(F1F3), Energy1750, f0max, Quantile1, Energy3750, SDF0, Energy3500. The data in Fig 1B shows an overall accuracy of 52.1%. Neutral voices' probability of being correctly predicted amounts to 44.1%, while probability of correctly predicting negative voices is 43.2% and positive voices is 48.5%, meaning all categorizations are above chance level (33.3%). Negative and positive vocal attitude respectively have a probability of being misclassified and attributed to neutral class label of 33.3% and 15.9%. The three most important features for valence classifications were f0min, D$f$ (F1F3), and Energy1750.

In conclusion, the SVM classification demonstrated that vocal profiles inferred at zero-acquaintance could be classified automatically from the acoustical parameters with a better than chance accuracy (chance level is 20% in the cross-dimensional classification, and 33.3% in the classification within dominance and valence). This suggests that some vocal profiles are acoustically distinctive, and that this distinction may partly be the base for trait inferences of valence and dominance.

**Random forests.** Random forests are applied as a comparison to SVM results and in order to reduce the risk of overfitting. Random Forests are a type of ensemble method which uses multiple decision trees based on random selection of variables and data to increase predictive performance of the algorithm. Thus Random Forests grow many trees and provide classes of dependent variables based on class votes obtained from each decision tree. Learning is achieved by recursively bootstrapping a different sample (training set) from the original data every time a tree is created. For each tree a random set of about 2/3$^{rd}$ or the dataset is used as a training set, while the remaining 1/3$^{rd}$ is used as a test set to determine the misclassification rate (or out-of-bag estimate). Model performance is obtained by considering each tree's

classification ("votes") and choosing the most popular. Random Forests are considered highly unbiased, and are particularly well suited for small sample sizes. A classification model is performed using the "random Forest" package [106] in RStudio v.1.2.5019 [107]. Dependent variable is the class label "attitude" which comprises four levels: Highly dominant and Highly trustworthy (HTHD) stimuli, Highly dominant and untrustworthy (LTHD) stimuli, Low dominance and Highly trustworthy stimuli (HTLD), and Low dominance Low trustworthy (LDLT) stimuli. Number of predictors sampled for splitting at each node is set at 9 (mtry = 9), and the number of trees is set at 10000 (ntree = 10000) based on the best recognition rate.

## Results

**Model performance.** Out of robustness concerns, we report both the permuting Out-of-bag (OOB) error and the Mean Decrease Gini as an indicator of the variable's impurity. The out-of-bag estimate computes the prediction error for every single tree. Ultimate OOB estimate of error rate after varying the number of trees is 36.51% (Table 2A), with forty-one incorrectly classified items out of ninety. Non-dominant and untrustworthy vocal profiles were classified the best, with a classification error of 0.14, while dominant and trustworthy were slightly misclassified (error: 0.31). Trustworthy but non-dominant vocal profiles failed at being classified correctly (error: 1). Local variable importance is indicated by the mean decrease Gini (S4 Fig in S1 File), the sum of all decreases in Gini impurity when a given variable is used to form a split at a node in the Random Forest, divided by the number of trees. Thus, a higher Mean Decrease Gini value indicates a more important variable.

Coefficients suggest minimum intensity levels (minIntensity), Energy500 and HNR are the three most important variables to discriminate between class labels. Hence in this classification, high levels of HNR are more likely to be labelled as untrustworthy/non-dominant (LTLD) when levels of energy in the 250-750Hz band are high and minimum intensity levels are low (Fig 3). Voices are labelled as Trustworthy-dominant (HTHD) for high levels of minimum intensity, combined with low HNR. In order to check if some features would be less-than-chance predictors of the vocal attitudes, we added noise by the mean of three random variables of different distributions (gaussian, uniform and binomial) to control for

**Table 2. RF Confusion matrices for the four vocal profiles.**

**(A) Full features classification**

|  |  | Predicted class | | | | Classification Error |
|---|---|---|---|---|---|---|
|  |  | HTHD | HTLD | LTHD | LTLD |  |
| Actual class | HTHD | **13** | 0 | 2 | 4 | 0.32 |
|  | HTLD | 0 | **0** | 0 | 7 | 1 |
|  | LTHD | 3 | 0 | **4** | 3 | 0.60 |
|  | LTLD | 3 | 1 | 0 | **23** | 0.15 |

**(B) Subset classification**

|  |  | Predicted class | | | | Classification Error |
|---|---|---|---|---|---|---|
|  |  | HTHD | HTLD | LTHD | LTLD |  |
| Actual class | HTHD | **11** | 0 | 2 | 6 | 0.42 |
|  | HTLD | 0 | **1** | 0 | 6 | 0.86 |
|  | LTHD | 3 | 0 | **4** | 3 | 0.60 |
|  | LTLD | 2 | 0 | 0 | **25** | 0.07 |

*Note.* (A) OOB estimate of error rate is 36.51% (mtry = 9, ntree = 10000). (B) OOB estimate of error rate is 34.92%. ntree = 6000. The number of variables randomly sampled at each split is adapted to the number of total variables (mtry = 3).

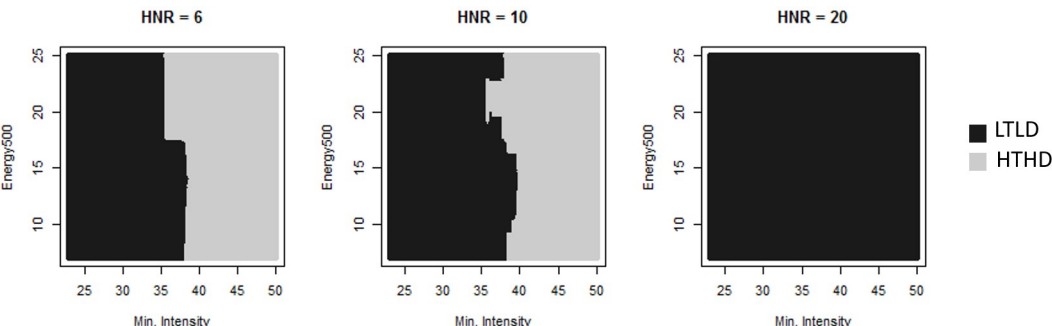

**Fig 3. Visualization of all decision trees in the RF, for energy levels in the 250-750Hz band, relative to minimum intensity and HNR.** Each dot indicates a decision tree's classification. HTLD and LTHD do not appear, indicating the combination of the three features were not significantly associated to those classes.

randomness in the features. The added noise were part of the least informative stratum (Gini coefficients $\leq 1.46$), suggesting that all used variables above that coefficient had a meaningful—however weak or strong—impact on the classification.

In random forests, the number of features can increase the generalization error, especially if there is redundancy. Hence a second classifier is trained, using a subset of twelve of the most important initial features (see Table 2B). We observed a slightly smaller OOB estimate of error rate than the "full featured" classifier (34.92%). In this second classification, trustworthy but non-dominant voices (HTLD) are still the most difficult samples to label. It is important to note that compared to the "full featured" classifier, labelling of trustworthy and dominant voices (HTHD) is less accurate.

## Conclusion

Automated classification using SVM showed greater than chances results to correctly classify vocal profiles using acoustical parameters. The highest level of accuracy was obtained for dominant and trustworthy profiles, followed by its polar opposite, non-dominant and untrustworthy profiles (see S4 Table in S1 File for a comparison in supporting material). Trustworthy but non-dominant voices were classified with the lowest accuracy, although still above chance levels. This class was the most misclassified, as it had a 20% chance to be wrongly labelled as "untrustworthy/non-dominant" (LTLD), but 60% to be correctly labelled. Above chance misclassification was observed for neutral valence-trustworthiness and neutral power-dominance, where the voices respectively had a 35.9% and 34.3% chance of being wrongly labelled as negative on the relative dimension. The random forest classification showed similar results, suggesting more contrasting acoustic substrates for trustworthy-dominant (HTHD) and untrustworthy/non-dominant (LTLD) profiles than for the profiles with mixed polarization on the two dimensions. The SVM classification showed that dominance classification returned the highest overall accuracy (58.6%) compared to valence (52.1%). The SVM analysis shows that feature selection based on relative feature weights can, to some extent, improve the classifiers' accuracy. Indeed, it is noteworthy that the best solution was not always the one with the most features, or only with the most important ones, but a combination of those. For instance, dominance classification dropped to 46.47% accuracy when only the three features with the most weight were kept for model training, and 55.56% when all features were kept. This suggests that complex interactions might be at play between acoustical features, irrespective of their individual position on the informativeness ranking. With this consideration, it is

noteworthy that some similarities were found in feature contribution in SVM and RF classifi-cations (see S4 Table in S1 File for comparisons). Indeed, SDintensity, Energy3250 and shim-mer contributed each time to classification of bi-dimensional vocal profiles.

## Discussion

Literature on impression formation suggests that first impressions are coherent, structured and core-centered [2, 18, 19, 108–110]. In the present study, we report that first impressions from vocal stimuli are indeed formed on a bi-dimensional space revolving around two main dimensions: perception of valence (with high loadings mainly on trustworthiness) and percep-tion of power (with high loadings mainly on dominance). This first result is consistent with lit-erature [2, 6, 8, 9, 13, 27, 111], and gives further support to McAleer et al.'s [13] results by verifying the structure of impression on longer vocal excepts. This methodology allowed for the examination of the role acoustical features relevant to person perception that are best mea-sured on longer durations. For instance, the implication of shimmers in valence perception. Interestingly, perception of male and female's dominance was not based on the same acoustical predictors but it is also possible that it relies strongly on the chosen rating scales. The rating scales from McAleer et al. [13] come from various sources relevant to impression formation, and appropriately represent the traits that hold supporting evidence for their importance in lit-erature. The measured traits vary widely in literature, providing small (or sometimes impor-tant) variations in semantic information provided by the interpretation of the voices. It is possible to consider that a third dimension would also explain (less importantly but still signif-icantly) how speakers are perceived. For instance, Sutherland et al. [25] investigated social inference from faces using both the rating scales found in Oosterhof & Todorov [5] (which supported a valence-trustworthiness and power-dominance space), and additional scales proven of importance in literature, namely "age", "babyfacedness", "health", "attractiveness". Their results showed that on top of the two expected dimensions, a third one revolving around "youthful-attractiveness" ratings allowed a better representation of the variance dataset once added to the model.

Overall, the results from machine learning analyses suggest that such methods are efficient in order to study voice and acoustic parameters, and the accuracy levels supports the advantage of a non-linear models in vocalization analyses. In contrast to earlier findings in emotional prosody recognition [95, 112], it is not surprising to find accuracies that do not go over the accuracy results in emotion identification, especially with such a small dataset. This can be imparted to the rather conceptual nature of attitudes. Indeed there is no clear consensus on the definition of a "trustworthy" voice for example, even less than for an "angry" voice, and the explicit categorization by a group of judges might also lead to variations in labelling, especially if categorization is implicit by nature. However a consensus can be found in receivers, as observed with the good judgment reliability. In this regard, error patterns in categorization (seen in confusion matrices) are a considerable source of information to better understand the nature of the inference process, by allowing to understand which labels are misattributed and how. Misclassification was lowest for bi-dimensional profiles, suggesting it is more accurate to consider the two dimensions combined rather than as independent factors of impression. However, further testing with bigger datasets would be needed to support this claim, as it is possible that this advantage partially reflects the extreme polarization chosen for the bi-dimen-sional profiles, while neutrals were considered for valence-trustworthiness and power-domi-nance profiles. Careful attention to data changes due to concept drifts would be needed in this situation [113].

In the first study, regression results of acoustical parameters shed light on the influence of constituents of voice quality (spectral balance, as well as energy distribution and variability) on person perception. In the second study, bi-dimensional classification was achieved by combinations of F0 parameters (F0 mean, F0 median, quantile 3, quantile 4), short-term variation of intensity, spectral balance parameters (PE1000, Hammarberg index, H1H2, CPP), shimmers and energy distribution in the 500-1000Hz and 3000-3500Hz bands. There is ample evidence that F0 parameters are amongst the most relevant features in affective vocalizations. The underlying assumption is that as pitch goes up with muscular tension [114], some high-activation emotional states such as hot anger can be associated to higher-pitch than low-activations ones such as sadness or boredom [95, 115, 116]. In the field of impression formation, Morton [48] compared F0 modulation to piloerection mechanisms, proposing that displaying a lower F0 aims at increasing the signaler's apparent size and threat potential. As such, F0 has been found to be used as a cue to physical strength despite not being related to actual strength [17, 72, 117]. However, the RF's classification of trustworthy/dominant and untrustworthy/non-dominant vocal profiles was mainly reflected by intensity parameters, vocal roughness (HNR, Shimmer), and energy distribution. The association with HNR might be tied to perception of age [79], which has been found to influence trustworthiness and dominance ratings [38]. Besides, energy distribution was initially proposed as a correlate of vocal arousal, under the assumption that activation of the ergotropic system (sympathetic arousal) along with emotions would be characterized by increased energy peaks in the higher frequency range [118]. Its importance in both classifiers suggests that it might also prove relevant for impression formation. Overall, the reported complex combinations of parameters to classify vocal profiles gives support to the consideration of complex parameter interactions, and challenges a one-to-one relationship between vocal features and perceived attitudes. The SVM classification also cast light on different combinations of features reflecting the power-dominance and valence-trustworthiness dimensions. Indeed, while some F0 (F0 max, SD) and energy parameters were used by both classifiers, dominance was specifically reflected by speech rate, intensity (Min, Mean), spectral balance parameters (CPP, D$f$(F1F5)), and vocal aperiodicity (HNR, shimmers, jitters). As a measure of periodicity, CPP correlates with breathiness and vocal roughness, and is inversely related to shimmers [119, 120]. Formant dispersion is related to perception of vocal depth (or resonance), and corresponds to a vocalizer's timbre. As a correlate of body size [53], there is considerable evidence for its use as a cue to physical and social dominance [17].

Meanwhile, apart from formant dispersion (D$f$(F1F3)), valence was mainly reflected by F0 parameters (Min, Max, SD, quantile 1). While formant dispersion and mean F0 largely rely on anatomical substrates, variability of F0 (SD-F0) may not be explained simply by size and shape of the vocal apparatus. A greater variability is perceived as a less monotone voice. Thus, in a study showing sex differences in SD-F0 and dominance perception, it has been suggested that SD-F0 could be affected by neuropsychological processes, and serve as an affiliation signal much like smiling [55]. In the same line, high-ranking members in a social hierarchy have been found to display less F0 variability than individuals with less power [121]. The present results support the need for further investigation of SD-F0's role on impression formation, relative to sex and the evaluative dimension. Despite some similarities, the differences in acoustic input feature informativeness for each dimension suggests that distinct combinations of acoustical inputs might be particularly relevant for the evaluation of the valence-trustworthiness and power-dominance dimensions.

Moreover, it is important to note power-dominance categorization was more accurate based on the acoustic properties of the signal than valence-trustworthiness categorization. It is not clear why power-dominance profiles benefit from a better categorization. In the first and second experiments, both acoustical results and Machine Learning analyses highlight the

relevance of acoustical features previously found to be implicated in evaluating arousal in emotional voices. Indeed, in both study 1 and 2, HNR, intensity and energy appear to be the most important cues for attitudinal categorization. One possible explanation might come from an imbalanced informative value between the two types of signals. In a lexical-based research on meaning of emotional words, Scherer and Fontaine [122] investigated the factorial structure of emotional meanings for faces, voices and body languages. They found that vocal expressions were predictive of both power and arousal, while valence and novelty factors were less related to vocal information, suggesting that vocal expression was a more appropriate source of power and arousal information than face was [86, 123, 124]. This difference might be due to the dynamic nature of voice, which makes arousal better carried through it. Arousal has been shown to be highly related to energy and spectral features, such as HNR, shimmers and intensity, as well as F0 levels [86, 95, 125–127]. HNR has also been linked to perception of aging [79] while energy and spectral measures have been extensively linked to speaker identification [128, 129], suggesting a possible importance of power-dominance related cues in speaker perception [86, 95]. Similarities between the acoustical cues usually implicated in arousal evaluations and the ones highlighted by both study 1 and 2 could reflect a dominance of features associated to physiological activation and strength in affective (emotional and attitudinal) voice evaluation. Another reason might be evolutionary. Although vocal attitudes might not share physiological activations found in emotion, it is sometimes posited that social evaluation consists in attuned perception of emotional counterparts [31, 130]. In previous literature on impression formation, valence-related percepts (communal content) were sometimes seen as more informative and more determinant of subsequent decisions for the receiver than power-related appraisal of others [30, 131–133]. However, those researches merely addressed facial evaluation, therefore dismissing the possibility of a specificity between inputs. The present accuracy findings, while preliminary, suggests voices seem to carry more discriminative cues related to power-dominance than valence-trustworthiness. Nonetheless, it is consistent with several claims in literature, that humans adapted for assessing physical strength from the voice as adaptive cue for fitness and survival [134]. In this regard, one might be surprised that pitch related cues (meanF0, minF0, maxF0, jitter) were not found to play a major role on impression formation in the present study, given that F0 has been tied to perception of—although not actual—physical strength [54] and dominance [60].

This study successfully shows that a bi-dimensional social space can be a valid frame to understand attitudinal rating to person perception, and first impressions. It re-affirms the ground for a valence-trustworthiness and power-dominance comprehension of rapid inference. The second experiment did not allow for a further investigation of this observation, as there was not enough observations in the categories of interest to train a model on, but it provides sufficient cause for further investigation of the functionality or theoretical bases of first impression formation from vocal cues. The ecological approach of facial impressions holds that first impressions have an adaptive role and result in approach or avoidance responses to stimuli [31]. Thus trustworthiness impression is based on resemblance to positive emotions or ecological signals (ex. baby face) that shows it is suitable to approach it, and dominance impression is based on cues resembling physical strength, or a person's capacity to harm [5]. In our results, dimensions for male voices were not the same as for female voices, suggesting an early influence of context in inference processes. There are several ways of explaining that observation. Firstly, expectations and conceptualization of female and male dominance differs in stereotype literature, such as female dominance is seen as more "socially oriented" and cooperative ("communal traits"), while male dominance is seen as more self-serving and competitive ("agentic traits") [135–138]. Moreover, expression of dominance can also differ between sex [139–141]. In a way, it lends weight to Asch's theory of person perception which

states that forming an impression requires a concept of a personality's "coherence" and "unity" [18, 19]. However most of the stereotype theories would imply that power-dominance is negatively associated with female competence and confidence instead of positively, contrary to our observations.

Other approaches of impression formation such as dual-process approaches suggest a two-step process. In the dual-process model of impression formation [142–144] a first assumption is made in an effortful process based on salient features, while a second process involving rule-based inference can also lead to impression formation. Fiske and Neuberg's [145] continuum model of impression formation assumes a category-based impression before using the target's individual attributes if motivation is high enough. The stereotype content model integrates consideration of group affiliation to social perception [2]. Although those models do not contradict the ecological approach, they assume a step-by-step process of person perception. Unfortunately, the present study does not allow to make a conclusion on the nature of this process because it does not directly address the unfolding of impression formation. However, it is interesting to note that McAleer's et al. [13] study did not report such gender effects from shorter speech extracts exposure. This could support the hypothesis of a step process of vocal impression, and uphold the realization of further investigations involving timing measures, such as gating methods already used for emotional and attitudinal prosody [146].

To add to this, independently of timing, the evaluative difference in attitudinal spaces between female and male speakers also raises the question of the role of social context and goals in first impression formation. Coincidentally, work addressing this effect has recently been emerging in literature. In the continuity of bi-dimensional approaches, Sutherland et al. [147] have argued for the possibility of an effect of social goals on person perception. Following up on this postulate, Collova, Sutherland and Rhodes [148] investigated dimensions of first impressions of children faces' in a series of studies. While bi-dimensional models rely on the assumption that impressions serve as a threat signal, the authors suggested that a functionalist approach would expect different evaluating objectives relative to the target (observed) population. For instance, social goals associated with children not being the same as for adults, dimensions of impression should not reflect perception of threat from children's faces. A first analysis revealed that a first dimension ("niceness") of children impression was conceptually close to the valence-trustworthiness evaluation, but that a second ("shyness") did not relate to the otherwise "dominance" dimension normally observed in adults. An implication of this is the possibility that impression formation is flexible, adaptable and context-dependent. Further research should take into account the social underpinnings in impression formation.

Partly because of its exploratory nature, there are some limitations to the present work. For instance, the categorization of vocal profiles from continuous measures needed for labelling in the classification lead to a severely reduced the dataset's size. The consequence was two-folds: first, it lead to inequal categories, which can impair the classifiers performance, and second it hampered any comparison of gender effects between machine learning and regression models. Larger datasets should be acquired to further test vocal classification according to the rules of art. Additionally, the present experiment used speech as stimuli. Vowels manifest through energy peaks (formants) at different frequencies, and each is formed by a different resonance production (produced by vocal tract shape). It is possible that speech articulation influenced the acoustical parameter's values, thus mistakenly leading to an over-representation of phonetical-dependent features in feature contribution measures. Thus, energy peaks in the spectrum should be interpreted with caution and further investigation should control for the effect of articulation on acoustical parameters, either by using non-speech vocalizations or filtering the signal.

To our knowledge, the question of social goals has never been addressed in vocal impression research. Moreover, there is still no previous evidence of a different perception of dominance in female and male speakers in first impression formation. This observation combined to the accuracy results obtained in the second experiment support a further exploration of the importance of voice for power-dominance perception, the unfolding of vocal impression and the effect of social context. Finally, the comparison between linear combinations of variables in the PCA and non-linear data-driven methods yields encouraging results for vocal impression analysis. It allows a more refined grasp on the dynamic interplay between acoustic features of the vocal signal, and establishes an acoustical ground for attitudinal labelling. In natural conversation, neutral tone prevails, especially amongst strangers [88], however it should not be taken for granted that it is devoid of any social or affective meaning. Most neutral conversations hold a social purpose. In evidence of the present results, we suggest that attitudinal prosody and its social context should be more investigated both in speech recognition and impression formation domains. The fundamental problem of understanding vocal impressions and relative decision-making is linked to the complex nature of the acoustical properties in voices, a better understanding of these snap judgments could improve models of human-machine interactions, synthetic speech, speech recognition, and many more.

## Supporting information

**S1 File.**
(DOCX)

## Author Contributions

**Conceptualization:** Cyrielle Chappuis, Didier Grandjean.

**Formal analysis:** Cyrielle Chappuis.

**Funding acquisition:** Didier Grandjean.

**Investigation:** Cyrielle Chappuis, Didier Grandjean.

**Methodology:** Cyrielle Chappuis, Didier Grandjean.

**Project administration:** Cyrielle Chappuis, Didier Grandjean.

**Resources:** Cyrielle Chappuis.

**Supervision:** Didier Grandjean.

**Validation:** Cyrielle Chappuis, Didier Grandjean.

**Visualization:** Cyrielle Chappuis.

**Writing – original draft:** Cyrielle Chappuis.

**Writing – review & editing:** Cyrielle Chappuis, Didier Grandjean.

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
