## [Decision Letter · Decision Letter 0]

19 Apr 2021

PONE-D-21-09962

Set the tone: first impressions from voices and affective classification using supervised machine learning techniques.

PLOS ONE

Dear Dr. chappuis,

Thank you for submitting your manuscript to PLOS ONE. After careful consideration, we feel that it has merit but does not fully meet PLOS ONE’s publication criteria as it currently stands. Therefore, we invite you to submit a revised version of the manuscript that addresses the points raised during the review process.

Based on the comments received from the reviewers and my own observation, I suggest major revisions for the paper.

We look forward to receiving your revised manuscript.

Kind regards,

Thippa Reddy Gadekallu

Academic Editor

PLOS ONE

Journal Requirements:

2. Please consider changing the title so as to meet our title format requirement (https://journals.plos.org/plosone/s/submission-guidelines). In particular, the title should be "Specific, descriptive, concise, and comprehensible to readers outside the field" and in this case it is not informative and specific about your study's scope and methodology.

3. Please improve statistical reporting and refer to p-values as "p<.001" instead of "p=.000". Our statistical reporting guidelines are available at https://journals.plos.org/plosone/s/submission-guidelines#loc-statistical-reporting.

 [The funders had no role in study design, data collection and analysis, decision to publish, or preparation of the manuscript.].

Reviewers' comments:

Reviewer's Responses to Questions

**Comments to the Author**

1. Is the manuscript technically sound, and do the data support the conclusions?

Reviewer #1: Yes

Reviewer #2: Partly

2. Has the statistical analysis been performed appropriately and rigorously? 

Reviewer #1: Yes

Reviewer #2: Yes

3. Have the authors made all data underlying the findings in their manuscript fully available?

Reviewer #1: Yes

Reviewer #2: Yes

4. Is the manuscript presented in an intelligible fashion and written in standard English?

Reviewer #1: Yes

Reviewer #2: No

5. Review Comments to the Author

Reviewer #1: The authors aims at extending the line of research on the acoustical bases of vocal impression in two ways: first (Study 1) by establishing a correspondence between acoustics and voice-first-impression (VFI) dimensions relative to sex, second (Study 2), by exploring the non-linear relationships between acoustical parameters and VFI by the means of machine learning models. This paper needs minor revision.

Below are my comments:

• The Introduction section should highlight the issues in concept drift. The contributions of the authors are not clear. They have mentioned in first contribution.

• There are some paragraphs that can be merged.

• We aim at extending the line of research on the acoustical bases of vocal impression in two ways: first (Study 1) by establishing a correspondence between acoustics and voice-first-impression (VFI) dimensions relative to sex, second (Study 2), by exploring the non-linear relationships between acoustical parameters and VFI by the means of machine learning models.? where is contribution and how it helps future research.

• similarly second and third contribution statement should be written in more focused way.

• writing is good, need to check the typo errors.

• paper is well-formatted, plz check the formatting of the reference

• I found some English mistakes please check them.

• why this specific Approach is applied for vocal impression, there is so many other statistical ways available. plz, justify it.

• please add some more related work related to PCA..

• Please cite the relevant literature but not limited to:

a) Bhattacharya, S., Maddikunta, P. K. R., Kaluri, R., Singh, S., Gadekallu, T. R., Alazab, M., & Tariq, U. (2020). A novel PCA-firefly based XGBoost classification model for intrusion detection in networks using GPU. Electronics, 9(2), 219.

b) Rehman, Z. U., Zia, M. S., Bojja, G. R., Yaqub, M., Jinchao, F., & Arshid, K. (2020). Texture based localization of a brain tumor from MR-images by using a machine learning approach. Medical hypotheses, 141, 109705.

c) Khan, H., Asghar, M. U., Asghar, M. Z., Srivastava, G., Maddikunta, P. K. R., & Gadekallu, T. R. (2021). Fake Review Classification Using Supervised Machine Learning. In Pattern Recognition. ICPR International Workshops and Challenges: Virtual Event, January 10–15, 2021, Proceedings, Part IV (pp. 269-288). Springer International Publishing.

d) Sai Ambati, L., Narukonda, K., Bojja, G. R., & Bishop, D. (2020). Factors Influencing the Adoption of Artificial Intelligence in Organizations-From an Employee's Perspective.

Reviewer #2: The work presented by author on Set the tone: first impressions from voices and affective classification using supervised

machine learning techniques needs major revisions.

1. Firstly, the work is of 78 pages!! which has irrelevant content as authors are deviating from the original research. It is advisable that authors should eliminate any unnecessary content under literature survey and add the data to support the literature.

2. Language and grammatical errors are at many places which needs to be corrected through out in the paper.

3. The work presented shows lack of clarity and it is not organized that any one who read the work understands it. The lack of clarity is because of unambiguous content and poor research methodology. It is advisable for authors to rephrase and redesign the methodology to support claims.

Lastly, the references are not cited properly and some references used at many places are not even in the scope of the research. Eliminate such references from the paper.

Authors are advised to cite below references in the paper

K. Chandra, G. Kapoor, R. Kohli and A. Gupta, "Improving software quality using machine learning," 2016 International Conference on Innovation and Challenges in Cyber Security (ICICCS-INBUSH), Greater Noida, India, 2016, pp. 115-118, doi: 10.1109/ICICCS.2016.7542340.

G. Arora, P. L. Pavani, R. Kohli and V. Bibhu, "Multimodal biometrics for improvised security," 2016 International Conference on Innovation and Challenges in Cyber Security (ICICCS-INBUSH), Greater Noida, India, 2016, pp. 1-5, doi: 10.1109/ICICCS.2016.7542312.

6. PLOS authors have the option to publish the peer review history of their article (what does this mean?). If published, this will include your full peer review and any attached files.

Reviewer #1: No

Reviewer #2: No

---

## [Author Response · Author response to Decision Letter 0]

16 Oct 2021

Thank you for your review of our paper. We have answered each of your points below. 

1. The Introduction section should highlight the issues in concept drift. The contributions of the authors are not clear. They have mentioned in first contribution.

A mention to the concept drift has been added to the discussion, in the situation where the present data set was used again for further implementation. However, the present research uses a dataset and analyses gathered at one point in time, and the timing of the research does not span several years which would provoke impactful changes in the dataset. Hence, we decided to keep this in the discussion. 

2. There are some paragraphs that can be merged.

In Study 1, stimuli preparation and personality measures were merged, and some of the text was added as a supporting information (S1 Text). Additional small changes were done throughout the text as to ensure that there was no overlap (e.g. in the “impressions” paragraph, we removed parts that were redundant with the conclusion). 

3. We aim at extending the line of research on the acoustical bases of vocal impression in two ways: first (Study 1) by establishing a correspondence between acoustics and voice-first-impression (VFI) dimensions relative to sex, second (Study 2), by exploring the non-linear relationships between acoustical parameters and VFI by the means of machine learning models.? where is contribution and how it helps future research.

Thank you for pointing this out. Extensive modifications have been done to emphasize the contribution, the abstract has been corrected for clarity and unnecessary parts of the introduction and discussion have been removed.

Study 1 contributes by replicating a study published in 2014 (McAleer, Todorov & Belin, 2014) on the subject of first impression formation. To date, investigation of the structure of voice impression formation is still scarce, and the present results provide support for the bi-dimensional perspective, on voices. 

Study 2 investigates the non-linear relationship between acoustical characteristics of voices in interaction with personality/attitudes attribution. To our best knowledge this has not been done before and literature. It successfully shows that: (1) automated classification can be used to determine perceived trustworthiness/dominance in novel voices; (2) non-linear relationships between acoustical parameters are crucial, which cannot be fully accounted for by linear regression models commonly used. On an applied perspective, it is important because it measures perceived trustworthiness and dominance in voice, which are dimensions that can be used to model synthetic speech and human-computer interaction. 

4. similarly second and third contribution statement should be written in more focused way.

We think this is an excellent suggestion and emphasized those contributions in the discussion. 

5. writing is good, need to check the typo errors.

Thank you for pointing this out. We proceeded to a second check and corrected the typos.

6. paper is well-formatted, plz check the formatting of the reference

Reference formatting has been checked and relevant corrections have been made.

7. I found some English mistakes please check them.

Thank you, as in point 5, we proceeded to a second check and corrected English mistakes.

8. why this specific Approach is applied for vocal impression, there is so many other statistical ways available. plz, justify it.

Thank you for your question. One of the prevalent classification method used for vocal emotion recognition in the literature is decision trees. The random forest, for instance, utilizes multiple randomly generated decision trees, which allows it to take advantage of all the benefits of decision trees, ensemble methods and bagging approaches. It extends on other bagging techniques, and exploits a random selection of the features. Another prevalent method is the Discriminant analysis. However, the SVM outperforms both discriminant analysis and logit regression (on bootstrapped samples): it provides accurate and robust classification even when inputs data are non-monotone and non-linearly separable (see Auria & Moro, 2008). In consequence, they enable to evaluate more relevant information in a convenient way. To date, it is a common method in emotional prosody classification (e.g.: Deng, Fruhholz, Zhang, Schuller, 2017), hence using it for perceived personality/attitudinal classification, which according to the emotion overgeneralization hypothesis is based on its resemblance to emotions, is a natural continuity. Hence out of consistency with literature on emotional prosody, we applied SVMs and RFs. 

9. please add some more related work related to PCA..

Thank you for this suggestion. We added a very important paper published this year which supports the use of such a methodology for first impression researches: 

Jones, B. C., DeBruine, L. M., Flake, J. K., Liuzza, M. T., Antfolk, J., Arinze, N. C., ... & Sirota, M. (2021). To which world regions does the valence–dominance model of social perception apply?. Nature human behaviour, 5(1), 159-169.

 As for the rest of the manuscript, a large number of the sources cited here use PCA methodology. Indeed, the present experiment partially replicates McAleer, Todorov and Belin’s (2014) study on voice and first impression formation, which is described extensively throughout the paper and also uses PCA. Many other sources use PCA methodology, as common in social cognition studies, such as Oosterhof and Todorov (2008), or Sutherland et al. (2013). 

10. Please cite the relevant literature but not limited to:

a) Bhattacharya, S., Maddikunta, P. K. R., Kaluri, R., Singh, S., Gadekallu, T. R., Alazab, M., & Tariq, U. (2020). A novel PCA-firefly based XGBoost classification model for intrusion detection in networks using GPU. Electronics, 9(2), 219.

b) Rehman, Z. U., Zia, M. S., Bojja, G. R., Yaqub, M., Jinchao, F., & Arshid, K. (2020). Texture based localization of a brain tumor from MR-images by using a machine learning approach. Medical hypotheses, 141, 109705.

c) Khan, H., Asghar, M. U., Asghar, M. Z., Srivastava, G., Maddikunta, P. K. R., & Gadekallu, T. R. (2021). Fake Review Classification Using Supervised Machine Learning. In Pattern Recognition. ICPR International Workshops and Challenges: Virtual Event, January 10–15, 2021, Proceedings, Part IV (pp. 269-288). Springer International Publishing.

d) Sai Ambati, L., Narukonda, K., Bojja, G. R., & Bishop, D. (2020). Factors Influencing the Adoption of Artificial Intelligence in Organizations-From an Employee's Perspective.

Thank you for suggesting this literature. Reference b has been added, however, the present paper presenting a social cognition research we estimate that it would be confusing to add GPU and network studies (a), or a pattern recognition study (c) in the literature. While they are relevant to the methodology of the research, we feel that they do not improve the understanding of the present research. 

Response to Reviewer 2:

Thank you for your comments. Our answers to your points are as follows.

1. Firstly, the work is of 78 pages!! which has irrelevant content as authors are deviating from the original research. It is advisable that authors should eliminate any unnecessary content under literature survey and add the data to support the literature.

Thank you for pointing this out. We agree with the reviewer’s assessment. Accordingly, in order to be more concise, a considerable reorganization and rewriting has been done.

Specifically:

- Paper is now 38 pages (without references and supporting information). As it contains two successive studies and integrates extensive statistical analyses, we think the length is justified.

- Introduction shortening (from 12 pages down to 7): we focused on the goals of the present study which are understanding the structure of impression formation, and the influence of acoustical parameters on impression formation. Thus, we removed the paragraphs on evolutionary bases (adaptive process) and integrated the vocal impressions paragraph and the personality profiles paragraph into a general introduction. 

- Language has been adapted as to be more straightforward

- Unnecessary content has been moved to the supporting information (e.g. Acoustic measures in Study 1, method). 

- Abstract has been rewritten (intro+conclusion)

2. Language and grammatical errors are at many places which needs to be corrected through out in the paper.

The paper has been checked and corrected for language and typos. 

3. The work presented shows lack of clarity and it is not organized that any one who read the work understands it. The lack of clarity is because of unambiguous content and poor research methodology. It is advisable for authors to rephrase and redesign the methodology to support claims.

Thank you for your feedback. Multiple adjustements have been done to simplify this paper. More focused title, abstract, introduction, and results shed light on the aims of the present research. Such clarifications make the purposes of the methodology more salient: such as the use of overt judgements in Study 1, followed by automated classification in Study 2. 

4. Lastly, the references are not cited properly and some references used at many places are not even in the scope of the research. Eliminate such references from the paper.

Thank you for pointing this out. Errors have been corrected. However, we could not determine which references are not in the scope or the research as noted above. Please let us know what references are problematic if they still appear in the paper after revision. 

Authors are advised to cite below references in the paper

K. Chandra, G. Kapoor, R. Kohli and A. Gupta, "Improving software quality using machine learning," 2016 International Conference on Innovation and Challenges in Cyber Security (ICICCS-INBUSH), Greater Noida, India, 2016, pp. 115-118, doi: 10.1109/ICICCS.2016.7542340.

G. Arora, P. L. Pavani, R. Kohli and V. Bibhu, "Multimodal biometrics for improvised security," 2016 International Conference on Innovation and Challenges in Cyber Security (ICICCS-INBUSH), Greater Noida, India, 2016, pp. 1-5, doi: 10.1109/ICICCS.2016.7542312.

Thank you for your insight, the references cited above provide an interesting perspective to the scope of the present work. We respectfully suggest not to add these, as they are focused on cyber security and the present paper is in the scope of social cognition. We fear that adding such references would be confusing to the readers.

---

## [Decision Letter · Decision Letter 1]

10 Feb 2022

PONE-D-21-09962R1

Set the tone: Trustworthy and dominant novel voices classification using explicit judgement and machine learning techniques

PLOS ONE

Dear Dr. chappuis,

Thank you for submitting your manuscript to PLOS ONE. After careful consideration, we feel that it has merit but does not fully meet PLOS ONE’s publication criteria as it currently stands. Therefore, we invite you to submit a revised version of the manuscript that addresses the points raised during the review process.

We look forward to receiving your revised manuscript.

Kind regards,

Felix Albu, Ph.D.

Academic Editor

PLOS ONE

Additional Editor Comments:

The authors should address the new comments.

Reviewers' comments:

Reviewer's Responses to Questions

**Comments to the Author**

1. If the authors have adequately addressed your comments raised in a previous round of review and you feel that this manuscript is now acceptable for publication, you may indicate that here to bypass the “Comments to the Author” section, enter your conflict of interest statement in the “Confidential to Editor” section, and submit your "Accept" recommendation.

Reviewer #1: All comments have been addressed

Reviewer #3: All comments have been addressed

Reviewer #4: All comments have been addressed

2. Is the manuscript technically sound, and do the data support the conclusions?

Reviewer #1: Yes

Reviewer #3: No

Reviewer #4: Yes

3. Has the statistical analysis been performed appropriately and rigorously? 

Reviewer #1: Yes

Reviewer #3: Yes

Reviewer #4: Yes

4. Have the authors made all data underlying the findings in their manuscript fully available?

Reviewer #1: Yes

Reviewer #3: No

Reviewer #4: Yes

5. Is the manuscript presented in an intelligible fashion and written in standard English?

Reviewer #1: Yes

Reviewer #3: Yes

Reviewer #4: Yes

6. Review Comments to the Author

Reviewer #1: Good Job! Thanks for addressing the revision comments. Please make sure you have aligned the references to meet the journal requirements.

Reviewer #3: I was not involved in the first round of review. But from the response to reviewer comments, I can see that authors taken maximum care in improving the article. I have the following concerns regarding the data and experiments with machine learning:

1) You have mentioned that Forty eight native French speakers were recruited and each speaker produced 4 sentences, which should result in 48*4= 192 sentences. But you have mentioned that 160 vocal stimuli in acoustic measures section. How come this number, please check.

2) Machine learning experiments: Experiments are conducted with K-fold validation. Does the authors ensured the training and testing does not consists same speaker data? If training consists of some part of the data of one speaker and testing consists of remaining data of the speaker, results may show higher accuracy due to speaker dependency. Strictly speaking, authors should conduct LOSO (leave one speaker out) cross-validation for reliability of the results as the amount is small. Please follow any standard paper for the same.

3) Authors should present the results (performance metrics) systematically for both SVM and RF classifiers. Presently, the results are not discussed coherently in both.

4) Authors should report the mean accuracy and standard deviation in accuracy as they are conducting experiments in K-fold validation (or LOSO). This will the show the reliability of the results.

5) Also authors should report the confusion matrices for both classifiers.

For your reference, I am giving one paper for reporting the results systematically: Glottal features for classification of phonation type from speech and neck surface accelerometer signals, Computer Speech & Language, 2021.

Otherwise, follow anyone of the standard paper which has studies on smaller amount of data.

Reviewer #4: Thanks for the authors for writing this manuscript. After related revisions, The manuscript can be accepted.

7. PLOS authors have the option to publish the peer review history of their article (what does this mean?). If published, this will include your full peer review and any attached files.

Reviewer #1: No

Reviewer #3: No

Reviewer #4: No

---

## [Author Response · Author response to Decision Letter 1]

27 Mar 2022

Dear reviewers,

We thank you for the time and expert attention given to our publication. We appreciate your insights and questions, which we discuss and attend to in the sections below. We are confident these adjustments improved the publication, and we hope they will meet the editor’s standards. 

Please find enclosed below the responses to the reviewers comments.

Sincerely,

Cyrielle Chappuis 

6. Review Comments to the Author

Reviewer #1: Good Job! Thanks for addressing the revision comments. Please make sure you have aligned the references to meet the journal requirements.

Reviewer #3: I was not involved in the first round of review. But from the response to reviewer comments, I can see that authors taken maximum care in improving the article. I have the following concerns regarding the data and experiments with machine learning:

1) You have mentioned that Forty eight native French speakers were recruited and each speaker produced 4 sentences, which should result in 48*4= 192 sentences. But you have mentioned that 160 vocal stimuli in acoustic measures section. How come this number, please check.

Thank you for bringing this to our attention. Indeed this is an oversight on our side, as we failed to mention that eight participants had been excluded from analyses due to faulty recordings. This has now been corrected.

“Forty-eight native French speakers between 18 and 35 years old (20F/20M, M = 22.47, SD = 2.61) were recruited at the University of Geneva, out of which eight participants whose recordings failed were excluded from the study »

2) Machine learning experiments: Experiments are conducted with K-fold validation. Does the authors ensured the training and testing does not consists same speaker data? If training consists of some part of the data of one speaker and testing consists of remaining data of the speaker, results may show higher accuracy due to speaker dependency. Strictly speaking, authors should conduct LOSO (leave one speaker out) cross-validation for reliability of the results as the amount is small. Please follow any standard paper for the same.

Thank you for your question and agree with the importance of this step in the classification analysis. We confirm that the SVM and RF were trained (training data) on the acoustic features of a subset of the vocal stimuli, and then were tested (testing data) to classify a new set of vocal stimuli into the possible categories. Hence, different stimuli are assessed in the training and testing. 

We followed Prof. Fruhholz’s procedure for emotional voice classification as a standard (https://www.biorxiv.org/content/10.1101/2020.05.04.076463v1.abstract) 

This step is reported page 23-24: “The k-fold cross validation parameter was set at 5, so the training set comprises four-fifth of the data ( 50 items) while one-fifth (13 items) is held out as testing set for each of all five experiments. Cross-validation solves the problem created by division of the dataset into a training set and a testing set by splitting the dataset into k-numbers of testing sets, subsequently fitting the model on all data, and finally computing the scores for number k of times. »

3) Authors should present the results (performance metrics) systematically for both SVM and RF classifiers. Presently, the results are not discussed coherently in both.

We agree that the results should be presented in a coherent way in both tested classifiers, as it is paramount to the good understanding of the paper. We were very attentive to deliver the results in the most clear and well reasoned way, for both classifiers. Indeed for both classifiers, we analysed the exact same categories for both classifiers, we used the same acoustical data, and we reported accuracy for the entire classifier as well as categories. Additionnally, we identified the most important acoustical features in both classifiers when they reached their best accuracy, following a comparative approach for both. We argue that this reasoning allows to drive sufficient insight for the conclusion, and that adding more discussion to the results could make the results more opaque to the reader.

4) Authors should report the mean accuracy and standard deviation in accuracy as they are conducting experiments in K-fold validation (or LOSO). This will the show the reliability of the results.

We took this point in careful consideration, and agree that when k-fold validation is used, it is good to present mean accuracy and standard deviation to account for the variability of the results. However, in the present protocol, we fixed the random part used in the k-fold in order to have a reproducible training. In consequence, the results we report are not based on several trainings but rather on one. 

5) Also authors should report the confusion matrices for both classifiers.

This point led to a careful revision of the present results. The confusion matrice for the first classifier (SVM) is reported in Figure 2, and the confusion matrice for the second classifier (RF) is reported as a table, in Table 3. 

For your reference, I am giving one paper for reporting the results systematically: Glottal features for classification of phonation type from speech and neck surface accelerometer signals, Computer Speech & Language, 2021.

Otherwise, follow anyone of the standard paper which has studies on smaller amount of data.

Reviewer #4: Thanks for the authors for writing this manuscript. After related revisions, The manuscript can be accepted.

---

## [Decision Letter · Decision Letter 2]

11 Apr 2022

Set the tone: Trustworthy and dominant novel voices classification using explicit judgement and machine learning techniques

PONE-D-21-09962R2

Dear Dr. chappuis,

We’re pleased to inform you that your manuscript has been judged scientifically suitable for publication and will be formally accepted for publication once it meets all outstanding technical requirements.

Kind regards,

Felix Albu, Ph.D.

Academic Editor

PLOS ONE

Additional Editor Comments (optional):

The decision for the paper is Accept.

Reviewers' comments:

Reviewer's Responses to Questions

**Comments to the Author**

1. If the authors have adequately addressed your comments raised in a previous round of review and you feel that this manuscript is now acceptable for publication, you may indicate that here to bypass the “Comments to the Author” section, enter your conflict of interest statement in the “Confidential to Editor” section, and submit your "Accept" recommendation.

Reviewer #1: All comments have been addressed

Reviewer #3: All comments have been addressed

2. Is the manuscript technically sound, and do the data support the conclusions?

Reviewer #1: Yes

Reviewer #3: Yes

3. Has the statistical analysis been performed appropriately and rigorously? 

Reviewer #1: Yes

Reviewer #3: Yes

4. Have the authors made all data underlying the findings in their manuscript fully available?

Reviewer #1: Yes

Reviewer #3: Yes

5. Is the manuscript presented in an intelligible fashion and written in standard English?

Reviewer #1: Yes

Reviewer #3: Yes

6. Review Comments to the Author

Reviewer #1: Good Job on addressing reviewer comments. Although not all suggested changes have been addrsssed but majority of those are included.

Reviewer #3: The authors have tried to address my comments and now the manuscript can be accepted. Authors should try to use the high resolution figures for clarity as some of the existing figures are very poor in quality. Also, try to make sure the captions of figures and tables are self-contained.

7. PLOS authors have the option to publish the peer review history of their article (what does this mean?). If published, this will include your full peer review and any attached files.

Reviewer #1: No

Reviewer #3: No

---

## [Editor Report · Acceptance letter]

20 Jun 2022

PONE-D-21-09962R2 

Set the tone: Trustworthy and dominant novel voices classification using explicit judgement and machine learning techniques 

Dear Dr. Chappuis:

I'm pleased to inform you that your manuscript has been deemed suitable for publication in PLOS ONE. Congratulations! Your manuscript is now with our production department. 

Kind regards, 

on behalf of

Dr. Felix Albu 

Academic Editor

PLOS ONE